# Hyphopodium-Specific Signaling Is Required for Plant Infection by *Verticillium dahliae*

**DOI:** 10.3390/jof9040484

**Published:** 2023-04-18

**Authors:** Qingyan Liu, Yingchao Li, Huawei Wu, Bosen Zhang, Chuanhui Liu, Yi Gao, Huishan Guo, Jianhua Zhao

**Affiliations:** 1State Key Laboratory of Plant Genomics, Institute of Microbiology, Chinese Academy of Sciences, Beijing 100101, China; 2CAS Center for Excellence in Biotic Interactions, University of Chinese Academy of Sciences, Beijing 100101, China; 3School of Life Sciences, Hebei University, Baoding 071000, China; 4Qilu Zhongke Academy of Modern Microbiology Technology, Jinan 250022, China

**Keywords:** fungal pathogens, *Verticillium dahliae*, colonization, hyphopodia, penetration peg

## Abstract

For successful colonization, fungal pathogens have evolved specialized infection structures to overcome the barriers present in host plants. The morphology of infection structures and pathogenic mechanisms are diverse according to host specificity. *Verticillium dahliae*, a soil-borne phytopathogenic fungus, generates hyphopodium with a penetration peg on cotton roots while developing appressoria, that are typically associated with leaf infection on lettuce and fiber flax roots. In this study, we isolated the pathogenic fungus, *V. dahliae* (Vda^Sm^), from *Verticillium* wilt eggplants and generated a GFP-labeled isolate to explore the colonization process of Vda^Sm^ on eggplants. We found that the formation of hyphopodium with penetration peg is crucial for the initial colonization of Vda^Sm^ on eggplant roots, indicating that the colonization processes on eggplant and cotton share a similar feature. Furthermore, we demonstrated that the VdNoxB/VdPls1-dependent Ca^2+^ elevation activating VdCrz1 signaling is a common genetic pathway to regulate infection-related development in *V. dahliae*. Our results indicated that VdNoxB/VdPls1-dependent pathway may be a desirable target to develop effective fungicides, to protect crops from *V. dahliae* infection by interrupting the formation of specialized infection structures.

## 1. Introduction

For any successful fungal-host interaction, including commensalism, symbiosis and pathogenesis, the most crucial event is the attachment and penetration of the plant surface [1]. During their colonization [1,2], fungi have evolved various strategies to overcome the barriers present in the host, such as by forming specialized infection structures [3,4,5,6] and secreting cell-wall-degrading enzymes [7]. A subset of fungal pathogens has evolved specialized infection structures to facilitate their penetration, whose infection strategies and morphology of the infection process have been well documented [1,2,3,5,8,9].

Appressoria have been thoroughly studied in *Magnaporthe oryzae*, which causes the most serious foliar fungal disease of cultivated rice [10]. Leaf infection by *M. oryzae* initiates from conidia that adhere to the leaf surface [10,11]. Conidia germinate and produce germ tubes that differentiate into heavily melanized penetration structures, known as appressoria. The appressoria then build up the tremendous turgor pressure to pierce the tough leaf surface [10,12]. It is noteworthy that *M. oryzae* also initiates root infection by forming the typical root pathogen hyphopodia, where the melanin layer is not observed. Moreover, root infection can lead to systemic invasion and classical disease symptoms on the aerial parts of the plant under laboratory conditions [10,12]. These results provide initial evidence for tissue-adapted fungal infection strategies [10]. In addition, a previous study showed that three *M. oryzae* genes essential for appressoria maturation in rice played only limited roles in the infection of *Arabidopsis*, suggesting that the pathogenic mechanisms are distinct from those in different hosts [13].

The infection structures of the soil-borne fungus *Verticillium dahliae*, which poses a major threat to more than 400 plant species by causing *Verticillium* wilt [14,15], are variable in different host species. In lettuce and fiber flax, *V. dahliae* developed an infection structure called appressoria at the junction of epidermal cells of host roots [16,17]. During the initial infection, conidia germinated on the root surface. Germinating hyphae grew parallel to the longitudinal axis of the root. Few hyphae developed appressoria along the junctions of root epidermal cells and penetrated an adjacent epidermal cell directly [16,17]. However, in *N. benthamiana*, hyphae invaded the host quickly through root wounds and the lateral root primordium without the formation of special infection structures [18]. When *V. dahliae* invaded oilseed rape [19] and sunflower [20], only slight hyphal swelling without penetration peg was observed before penetration. During oilseed rape infection, random growth of *V. dahliae* hyphae on the oilseed rape root surface was observed but not following any pattern [19]. For successfully colonizing the sunflower root, hyphae grow along the longitudinal grooves of epidermal cells, a process similar to that in lettuce. Additionally, abundant hyphae were observed at the protrusion sites of taproots, suggesting that these sites may facilitate *V. dahliae* sunflower root colonization [20]. For a cotton isolate of *V. dahliae*, strain V592, we incipiently observed slight hyphal swelling followed by a narrow penetration peg during infection of *Arabidopsis* roots [21]. Subsequently, the slight hyphal swelling cell was molecular, characterized as a typical infection structure called hyphopodium, which develops the penetration peg required for V592 to breach the cotton root cell wall during the initial colonization [22]. Hyphopodium-specific VdNoxB/VdPls1-mediated reactive oxygen species (ROS) production elevates Ca^2+^ accumulation in hyphopodia and then activates VdCrz1 signaling to form penetration pegs [22]. NADPH oxidase (Nox) is the major enzymatic producer of ROS, which has been shown to be crucial for fungal hyphal tip growth and fungal virulence [23,24,25,26,27]. Endogenous ROS elevation produced by Nox-activated Ca^2+^ channels facilitate Ca^2+^ influx for fungal cell polarity [28,29,30]. We found that ROS-Ca^2+^ signal plays a pivotal role in the fungal pathogenicity by regulating the penetration peg formation [22]. Furthermore, we demonstrated that the V592 infection structure not only functions as a colonization apparatus but also provides a unique interface for the secretion of fungal effectors [31]. Whether this hyphopodium molecular feature exists in other *V. dahliae* strains remains unknown.

In this study, we collected *Verticillium* wilt eggplants (*Solanum melongena* L.) from Hebei, China, and isolated the pathogenic fungus. Internal transcribed spacer (ITS) sequencing results indicated that the pathogenic fungus was *V. dahliae* (Vda^Sm^). The objectives of this study were to characterize the infection process of Vda^Sm^, determine whether Vda^Sm^ evolves infection structure, and identify the molecular features and the role of infection structure in the pathogenicity of Vda^Sm^.

## 2. Materials and Methods

### 2.1. Fungal Recovery, Culture Conditions, ITS Analysis and Infection Assays

The eggplants with *Verticillium* wilt symptoms were collected from a field in Guan County (39.43° N, 116.23° E), Langfang City, Hebei Province, China. The stems of the pathogenetic eggplants were placed in paper bags. The stems were cross-sectioned into 1 cm slices and soaked in 75% alcohol for approximately 10 min. Then, these slices were transferred to 30% sodium hypochlorite and soaked for approximately 20 min for surface sterilization. After rinsing three times with sterile water, the samples were cultured at 26 °C on potato dextrose agar (PDA) medium. Isolates were purified using single spore isolation [32]. Individual colonies were picked up and transferred to another PDA plate to continue growth for morphological observation or DNA extraction. For ITS analysis, fungal DNA extraction followed a previous description [33]. The ITS region was amplified with conserved primers [34] and sequenced by Suzhou Ribo Life Science Co., Ltd. (Ribo, Suzhou, China) (https://www.ribolia.com/en, accessed on 13 April 2023). Blastn (https://blast.ncbi.nlm.nih.gov/Blast.cgi, accessed on 13 April 2023) with default parameters was used to search the homologous sequences of ITS.

This isolate was stored at −80 °C and reactivated on PDA for pathogenicity assays. For the eggplant infection assays, the conidia were cultured in liquid Czapek–Dox medium, shaking at 200 rpm and 26 °C in the dark. Twelve-day-old seedlings of eggplants were irrigated with 1 × 10^7^ cfu/mL spores of Vda^Sm^ for infection. Disease progression in eggplants was recorded over time for at least 20 days. Disease symptom was classified into five grades: 0 (asymptomatic), 1 (0–25% leaf wilted or dropped off), 2 (25–50% leaves wilted or dropped off), 3 (50–75% leaves wilted or dropped off), and 4 (75–100% leaves wilted or dropped off or plant died).

### 2.2. Penetration Assays and Confocal Laser Scanning Microscopy (CLSM)

Minimal medium (MM) [22] was used for penetration assays. The Vda^Sm^ cultures were incubated on a cellophane membrane (DINGGUO, Beijing, China), which was overlaid onto MM. To determine if any Vda^Sm^ penetrated the cellophane, the hyphae were observed in the medium after removing the membranes. The experiments for each colony were repeated independently at least three times. For hyphopodium detection, the mycelium was grown on cellophane for 2 days and observed as previously described [22]. The protocol of plasma membrane staining using FM4-64 (ThermoFisher, Shanghai, China), ER-Tracker (ThermoFisher, Shanghai, China) staining and protein localization assays have been described [22,31].

To observe the infection process of Vda^Sm^, eggplant roots (Hang No. 1) were inoculated for 8 days and sectioned. Fluorescent photographs were captured using a Leica SP8 confocal laser scanning microscope system [21,22].

### 2.3. Construction and Transformation

To obtain GFP-labeled Vda^Sm^-GFP, the pNEO-olic GFP plasmid (stored in our laboratory) was transferred to Vda^Sm^. G418 was used to select the transformants on PDA medium.

To generate the knockout plasmids, pKOVsmNoxB, pKOVsmPls1 and pKOVsmCrz1, upstream and downstream genomic sequences of these genes were amplified with the corresponding primers (Appendix A). The paired sequences were inserted into a position flanking the hygromycin resistant cassette of the vector pGKO-HPT with the Exnase MultiS (Vazyme, Nanjing, China), and then the knock-out plasmids were transformed as previously described [35].

To produce the complemented strains, including VdΔ*noxb/VdNoxB* and VdΔ*pls1/VdPls1*, GFP-fused VdNoxB and VdPls1 under the native promoter were introduced into VdΔ*noxb* and VdΔ*pls1*, respectively. RFP-fused VdKar2 under the Tef promoter were introduced into VdΔ*noxb/VdNoxB* and VdΔ*pls1/VdPls1*, to detect the location of VdNoxB and VdPls1, respectively. The fusion plasmids were constructed as previously described [22], and the primers are listed in Appendix A.

### 2.4. Detection of ROS and Ca^2+^

For ROS detection, DAB staining solution (1 mg/mL, pH adjusted to 6.5–7.0) was prepared in PBS buffer. After culturing on cellophane membranes for 2 days, the colonies of Vda^Sm^ were cut and floated on the staining solution (1 mL). These colonies were cultured for 8 h in the dark at room temperature, rinsed twice and observed under a microscope.

Ca^2+^ detection followed our previous description [22].

### 2.5. Quantitative Real-Time PCR (qRT-PCR)

For qRT-PCR, total RNA was isolated from fungi using hot-phenol extraction [36]. gDNA wiper (Vazyme, Nanjing, China) was used to remove the residual DNA. Then, the RNA was reverse transcribed into cDNA using HiScript II Q RT Supermix (Vazyme, Nanjing, China). cDNA was subjected to qRT-PCR with a BioRad CFX96 Real-Time system using ChamQ SYBR qPCR MasterMix (Vazyme, Nanjing, China). β-Tubulin was included in the assay for normalization. The relative quantification was analyzed using the 2^−ΔΔCT^ method. For each sample, at least three biological replicates and three technical replicates were performed. The primers are listed in Appendix A.

### 2.6. Southern Blot

A total of 20 μg of genomic DNA was completely digested by proper restriction enzymes, and separated using agarose gel electrophoresis as previous described [22]. Gene-specific probes were amplified with primers listed in Appendix A and labeled with Biotin-11-dUTP (Thermo Fisher, Waltham, MA, USA, R0081). The chemiluminescence image analysis system (Tanon, St Andrew, UK, Tanon-4600SF) was used to detect the hybridization signals.

## 3. Results

### 3.1. Isolation and Identification of Pathogens Causing Verticillium Wilt in Eggplants

Based on plant symptoms, we collected diseased eggplants with the typical symptom of *Verticillium* wilt in the field. To isolate the pathogen, the stem sections of diseased plants were surface-sterilized and transferred to potato dextrose agar (PDA) medium (Figure 1a). The hyphae that grew around the tissue slices of eggplant were transferred to another PDA medium to continue growth (Figure 1a). In the laboratory, the fungus caused typical *Verticillium* wilt symptoms in healthy plants, such as leaf wilting and chlorosis and plant stunting (Figure 1b).

For molecular identification, the ITS region of the fungus was sequenced using the primer pairs ITS1/ITS4 [34]. Blastn analysis revealed that the ITS sequence was 99–100%, identical to the *V. dahliae* reference sequences. PCR assays, with two primer pairs which were designed to differentiate *Verticillium* species [37], further confirmed that the isolate is *V. dahliae* (Appendix A). Therefore, we conclude that the *V. dahliae* recovered from diseased eggplants (Vda^Sm^) was the pathogenic fungus causing *Verticillium* wilt.

### 3.2. Colonization Process on Eggplant Roots by GFP-Labeled Vda^Sm^

To explore the colonization process of Vda^Sm^, we generated a GFP-labeled isolate, Vda^Sm^-GFP. Green fluorescence was detected under a microscope to confirm the stable expression of GFP (Figure 2a). When cultured on PDA, both Vda^Sm^ and Vda^Sm^-GFP were similar in colony morphology and growth rate (Appendix A). Moreover, disease severity on eggplants caused by Vda^Sm^ and Vda^Sm^-GFP did not show any obvious difference (Figure 2b). Given that its pathogenicity was unaffected, Vda^Sm^-GFP was used to study the infection processes on eggplants.

The roots of infected eggplant were observed using the confocal laser scanning microscopy (CLSM). After 12 h inoculation (hpi), a small fraction of conidia germinated on the root surface at random sites (Figure 3a,b). Massive conidia that had germinated were observed at 24 hpi (Figure 3c). By 3 dpi, mycelium expanded, and hyphae covered the root surface. A fraction of hyphae expanded in parallel along the epidermal cells (Figure 3d). Only a few hyphae, tightly adhering to the surface, penetrate intercellularly into the epidermal cells (Figure 3e). At the site of penetration, we observed slight swelling of the elongating hyphae (Figure 3e). After successfully invading the root, the hyphae elongated parallelly along the longitudinal axis. The intercellular hyphal swelling between the epidermal cell junctions at the site of penetration to an adjacent cell was observed at 4 dpi (Figure 3f). By 6 dpi, the hyphae reached the vascular tissue and continued to grow and ramify, forming a hyphal net within the xylem vessels (Figure 3g). The root xylem vessels filled with hyphae were observed by 8 dpi (Figure 3h). Our results indicated that the invasion of eggplant roots by Vda^Sm^ shares a colonization process similar to that of V592 on cotton plants [21].

### 3.3. VdNoxB and VdPls1 Are Required for Penetration Peg Formation and Fungal Pathogenicity

We previously demonstrated that VdNoxB and VdPls1 from V592 are specifically expressed in infection structure and hyphopodium, and are indispensable for penetration peg formation in the colonization of V592 on cotton plants [22]. Therefore, to figure out whether the swelling hyphae (Figure 3e,f) were infection structures with specific molecular feature similar to V592 on cotton plants [22], we amplified and confirmed *VdNoxB* and *VdPls1* homologous sequences from Vda^Sm^. We then generated knockout mutants, VdΔ*noxb* and VdΔ*pls1*, using the homologous recombination method (Appendix A) [35]. There were no noticeable morphological differences between the Vda^Sm^ and the mutant strains (Figure 4a). GFP-fused VdNoxB and VdPls1 under the native promoter were introduced into VdΔ*noxb* and VdΔ*pls1* mutants, respectively, to produce the complemented strains VdΔ*noxb/VdNoxB* and VdΔ*pls1/VdPls1*. Green fluorescence was detected, confirming the stable expression of GFP-fused VdNoxB and VdPls1 in complemented strains (Appendix A).

We first examined the penetration abilities of the Vds^Sm^, VdΔ*noxb* and VdΔ*pls1* mutants and the complemented strains VdΔ*noxb/VdNoxB* and VdΔ*pls1/VdPls1* on a cellophane membrane laid on MM, which is used for V592 to induce hyphopodia and penetration pegs [22]. We observed that Vda^Sm^ and both complemented strains penetrated from the cellophane membrane and grew on the medium at 3 dpi (Figure 4b). However, fungal hyphae penetration from the cellophane membrane was not observed for either VdΔ*noxb* or VdΔ*pls1* mutants (Figure 4b), suggesting that VdNoxB and VdPls1 were also required for Vds^Sm^ to induce infection structure. Indeed, hyphopodia with clear penetration pegs to breach the cellophane membrane for Vda^Sm^ and complemented strains were observed under microscopy (Figure 4c). Even though hyphopodia was observed, neither the VdΔ*noxb* nor VdΔ*pls1* mutants generated penetration pegs (Figure 4c). These data demonstrate that VdNoxB and VdPls1 are indispensable for penetration peg formation in Vda^Sm^ to penetrate the cellophane membrane.

We then investigated the roles of VdNoxB and VdPls1 in the pathogenicity of Vda^Sm^ on eggplants. The eggplants were inoculated with spores from the wild-type Vda^Sm^, VdΔ*noxb* and VdΔ*pls1* mutant strains, as well as with the complemented strains, VdΔ*noxb/VdNoxB* and VdΔ*pls1/VdPls1*. Compared to Vda^Sm^, both the VdΔ*noxb* and VdΔ*pls1* mutant strains displayed significantly reduced disease severity in eggplants (Figure 5a,b). The loss of virulence was restored in both VdΔ*noxb*/Vd*NoxB* and VdΔ*pls1/VdPls1* strains (Figure 5a,b). Isolation of diverse Vda^Sm^ strains from the infected plants was used to confirm the successful inoculation (Figure 5c). These results indicate that VdNoxB and VdPls1 are essential to the pathogenicity of Vda^Sm^.

Similar to our previous studies on cotton [21,22], we conclude that special infection structures are required for *V. dahliae* strains colonization of various host plants. A conservative mechanism involving VdNoxB and VdPls1 would be indispensable for penetration peg formation.

### 3.4. VdNoxB and VdPls1 Were Highly Expressed in Hyphopodia and Localized at the Position of Penetration Peg Emergence in Vda^Sm^

NoxB and Pls1 were reported to be associated with the ER and specifically accumulate in the infection structures in *V. dahliae* [22], *Magnaporthe oryzae* [38] and *Botrytis cinerea* [39]. VdNoxB and VdPls1 of V592 particularly colocalized at the position of penetration peg emergence and VdPls1 is required for the plasma membrane localization and activation of VdNoxB [22]. To investigate the cellular localization of VdNoxB and VdPls1, the complemented strains VdΔ*noxb/VdNoxB* and VdΔ*pls1/VdPls1*, in which GFP-fused VdNoxB or VdPls1 was introduced into the corresponding mutant (Appendix A), were examined under CLSM. The functional activities of GFP-fused proteins were confirmed by complementation of both penetration ability (Figure 4) and pathogenicity (Figure 5). The hyphae grown on the cellophane membrane were stained by ER-Tracker Blue-White DPX (Figure 6a). Either GFP-fused protein colocalized with the ER-tracker signals (Figure 6a). Furthermore, the RFP was fused with the *V. dahliae* homolog of yeast KAR2, located at the ER and nuclear envelope [40]. RFP-fused VdKar2 was introduced into the VdΔ*noxb/VdNoxB* and VdΔ*pls1/VdPls1*. Red fluorescence was detected under a microscope, confirming the stable expression of VdKar2-GFP (Figure 6b). Similar to Blue-White DPX staining, overlapping of the RFP signal with the GFP fluorescence in both VdΔ*noxb/VdNoxB/VdKar2* and VdΔ*pls1/VdPls1/VdKar2* was also observed (Figure 6b). Notably, at the base of hyphopodia where penetration pegs developed, the plasma membrane of VdΔ*noxb/VdNoxB* and VdΔ*pls1/VdPls1* was clearly stained using FM4-64 (Figure 6c), where linescans also showed the strong GFP signals in a transverse section of individual hyphopodium (Figure 6d). Taken together, these results indicated that both VdNoxB and VdPls1 of Vda^Sm^ are associated with the ER, and localized at the position of penetration peg emergence in Vda^Sm^, which is similar to VdNoxB and VdPls1 location in V592 [22].

### 3.5. VdNoxB/VdPls1-Mediated ROS Production Coupled with Ca^2+^-Activated VdCrz1 Signaling in the Hyphopodium

In V592, VdPls1 regulates ROS burst by influencing the plasma membrane localization of VdNoxB. The VdNoxB/VdPls1-dependent ROS burst is essential for the free Ca^2+^ elevation in the hyphopodium, which activates VdCrz1 signaling to induce penetration peg development [22].

To confirm whether penetration peg formation in Vda^Sm^ shared a molecular mechanism similar to that in V592, we first detected the ROS burst in the hyphopodium. Vda^Sm^ hyphae grown on the cellophane membrane were treated with DAB staining for CLSM observation. Intensive ROS signals were detected at the base of hyphopodia in Vda^Sm^, where the penetration pegs were generated (Figure 7a). In contrast, no signal was detected for ROS-specific accumulation in either the VdΔ*noxb* or VdΔ*pls1* mutant strains (Figure 7a). Moreover, a tip-high Ca^2+^ gradient in the hyphopodia of Vda^Sm^ was observed with the intracellular calcium indicator Fluo-a AM, whereas Ca^2+^ was not detectable in the hyphopodia of the VdΔ*noxb* and VdΔ*pls1* strains (Figure 7b). Our data indicate that VdNoxB and VdPls1 are required for ROS production and Ca^2+^ elevation in the hyphopodium [22].

Next, we examined the expression of *VdCrz1* and its potential targets *VdLcc* and *VdRhom*, encoding *M. oryzae* orthologs of laccase and rhomboid family membrane protein, respectively [41], to confirm whether Ca^2+^ elevation affected VdCrz1 signaling in Vda^Sm^. The expression of *VdCrz1* was significantly reduced in VdΔ*noxb* and VdΔ*pls1* compared to wild-type Vda^Sm^ (Figure 7c). As expected, the two target genes of *VdCrz1* were also downregulated in both mutant strains (Figure 7c). We then generated the *VdCrz1* knockout mutant, VdΔ*crz1* (Appendix A), and assayed its ability to induce penetration peg formation and pathogenicity in eggplants. Compared to Vda^Sm^, VdΔ*crz1* did not exhibit obvious phenotypic differences on PDA plates (Figure 8a). Similar to VdΔ*noxb* and VdΔ*pls1*, VdΔ*crz1* hyphae penetration from the cellophane membrane at 3 dpi (Figure 8b) and the formation of a penetration peg at 2 dpi was not observed (Figure 8c). Pathogenicity assays showed that VdΔ*crz1* displayed greatly reduced virulence in eggplants compared to Vda^Sm^ (Figure 8d,f). Isolation of Vda^Sm^ and VdΔ*crz1* strains from the infected plants was used to confirm the successful inoculation (Figure 8e). Taken together, our results demonstrated that VdNoxB/VdPls1-mediated Ca^2+^ elevation activates VdCrz1 signaling to regulate the penetration ability and pathogenicity of Vda^Sm^.

## 4. Discussion

In this study, we demonstrated that the formation of hyphopodia and penetration pegs is required for the initial colonization of Vda^Sm^ on eggplant roots. VdNoxB and VdPls1 are indispensable for penetration peg formation and essential to the pathogenicity of Vda^Sm^. Both VdNoxB and VdPls1 localized at the base of hyphopodia, where the penetration pegs were generated. Additionally, VdNoxb and VdPls1 are required for the ROS production. Furthermore, the VdNoxB/VdPls1-dependent ROS burst elevates Ca^2+^ accumulation in the hyphopodia, which activates VdCrz1 signaling to regulate penetration peg formation (Figure 8g).

To enter the underlying host tissues, many fungi generate elaborate infection structures from emerging penetration hyphae to breach the cuticle and epidermal cell wall [1,42]. Interestingly, studies on host penetration by *Rhizoctonia solani* showed that the ability to generate infection structures is highly variable even within a given species [42,43]. Similarly, *V. dahliae* develops infection structures with different characteristics to adapt to host specificity. An appressoria, which is a requirement for leaf infection, was observed during the invasion of *V. dahliae* on lettuce and fiber flax roots [16,17]. However, we demonstrated that the formation of hyphopodia with penetration peg is essential for the initial colonization of V592 on cotton roots [22]. In this study, we isolated the *V. dahliae* strain Vda^Sm^ from *Verticillium* wilt eggplants (Figure 1) and observed its infection process with a GFP-labeled isolate (Figure 2). Hyphal swelling at the site of penetration was observed (Figure 3). The penetration ability assay showed that the hyphopodium with penetration pegs are indispensable to breach the cellophane membrane (Figure 4). Furthermore, we demonstrated that the hyphopodia and penetration pegs play essential roles in initial colonization and pathogenicity of Vda^Sm^ on eggplant (Figure 5Similar to our previous observation [21,22], we conclude that *V. dahliae* colonized eggplant, cotton and *Arabidopsis* with a similar process.

The tissue-adapted infection strategies of *M. oryzae* have clarified the difference in appressoria and hyphopodia [10,12,13,38]. On the leaf, *M. oryzae* develops heavily melanized appressoria associated with classical foliar infection, and then the appressoria build up tremendous turgor pressure to penetrate the tough surface [12,44,45]; on the root, hyphal swellings resembling the simple structure hyphopodia has been evident [12]. These results indicated that it is harder for the fungus to breach the leaf than to penetrate the root. Our previous study showed that the average diameter of *V. dahliae* hyphopodia was smaller than that of *M. oryzae* appressoria [31], which suggests that more pressure is needed for the fungus to breach the plant leaf compared to the root. Therefore, the previous findings of slight hyphal swelling during *V. dahliae* infection on oilseed rape and sunflower [19,20] that was thought as non-structural development without molecular characterization would be worth studying further.

Although there is considerable variation in morphology between appressoria and hyphopodia, a study showed that the infection structures share common genetic requirements during *M. oryzae* colonization [38]. In general, ROS generation is required for the differentiation of a penetration peg from appressoria [23,25,46] or hyphopodia [22]. Nox enzymes function as the major enzymatic producer of ROS [47,48,49]. In several fungal pathogens, the NoxA and NoxB proteins have been shown to be crucial for hyphal tip growth, tissue invasion and virulence [24,26,27,50]. ROS elevation, produced by NoxB, activates Ca^2+^ channels to facilitate Ca^2+^ influx for the plant root-hair cell and the fungal hyphae tip polarity [28,29,30]. Pls1, which is expressed during appressoria development, is presumed to be the corresponding integral membrane adaptor for the assembly of the NoxB complex [39,51]. In this present study, VdNoxB and VdPls1 of Vda^Sm^ were indispensable for penetration peg formation on the cellophane membrane (Figure 4). We observed that both VdNoxB and VdPls1 localize on the plasma membrane (Figure 6a), particularly at the position of penetration peg emergence (Figure 6b). Similar to our previous results on V592 [22], the ER and plasma membrane location of VdNoxB/VdPls1 was essential for penetration peg formation. In our previous study, yeast two-hybrid and bimolecular fluorescence complementation assays provided evidence of a direct physical interaction between VdNoxB and VdPls1 [22]. Furthermore, we demonstrated that the plasma membrane localization of VdNoxB and VdPls1 is required for the ROS burst in the hyphopodia (Figure 7a), which elevates Ca^2+^ accumulation at the base of hyphopodia (Figure 7b). Consistent with our previous results on V592 [22], VdNoxB/VdPls1-dependent Ca^2+^ elevation activated VdCrz1 signaling, and VdCrz1 and its targets were significantly reduced in VdΔ*noxb* and VdΔ*pls1* compared to Vda^Sm^ (Figure 7). Similar to VdΔ*noxb* and VdΔ*pls1*, penetration peg formation was not observed in VdΔ*crz1* on the cellophane membrane at 2 dpi (Figure 8b,c). VdΔ*crz1* displayed significantly reduced virulence in eggplants (Figure 8d).

## 5. Conclusions

In summary, we observed the infection process of Vda^Sm^ on eggplants and identified the infection structure and its molecular features. Similar to our previous study on V592 infection on cotton plants, we conclude that a common genetic pathway regulates host-specific infection-related development in the soil-borne fungus *V. dahliae*. The finding that Nox/Pls-dependent signaling is required for appressorium formation [23,51], suggests that Nox/Pls are key components for this pathogenic fungal colonization of their hosts. Therefore, Nox/Pls-dependent signaling maybe a desirable target for fungicides. The elaboration of regulatory mechanisms in the upstream of Nox/Pls-dependent signaling, such as cAMP [38,52], is worth studying further. A clear understanding of the process of colonization might help to develop effective fungicides for inhibiting the *V. dahliae* infection process.

## Figures and Tables

**Figure 1 jof-09-00484-f001:**
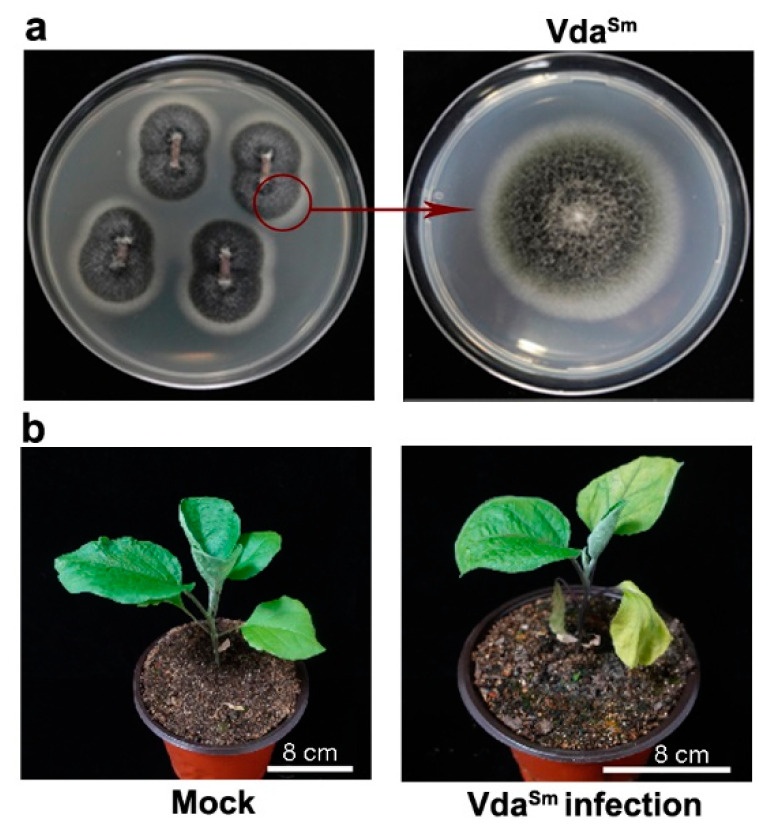
Eggplant wilt disease symptoms caused by infection with *V. dahliae* recovered from diseased plants. (**a**) Hyphae grew from two ends of the cut stems and were transferred to another PDA medium to continue growth; (**b**) Vda^Sm^ infection caused leaf wilting and chlorosis and plant stunting. Similar results were obtained from 20 plants, and representative photographs are shown.

**Figure 2 jof-09-00484-f002:**
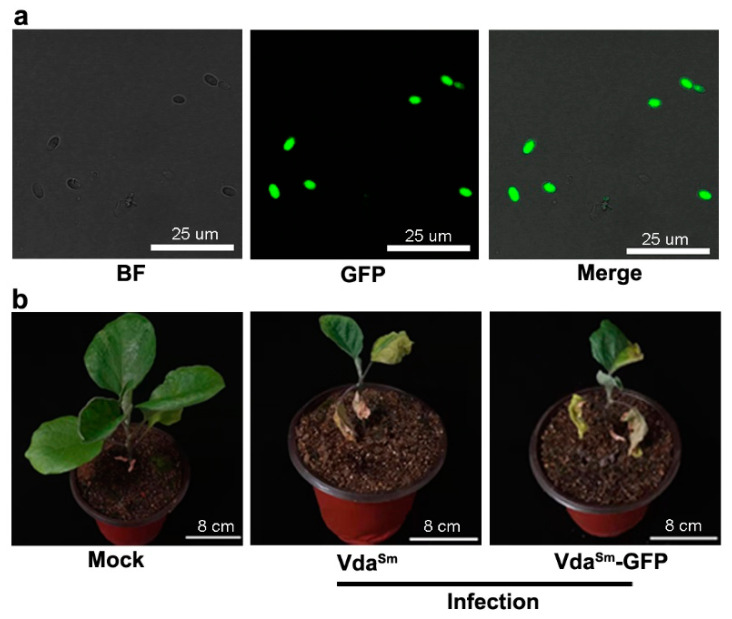
The pathogenicity caused by Vda^Sm^-GFP did not significantly differ from the wild-type isolate. (**a**) Confocal micrograph of green fluorescence by GFP-labeled Vda^Sm^; (**b**) Similar wilt symptoms were observed with Vda^Sm^ and Vda^Sm^-GFP on 20 eggplants, and representative photographs are shown.

**Figure 3 jof-09-00484-f003:**
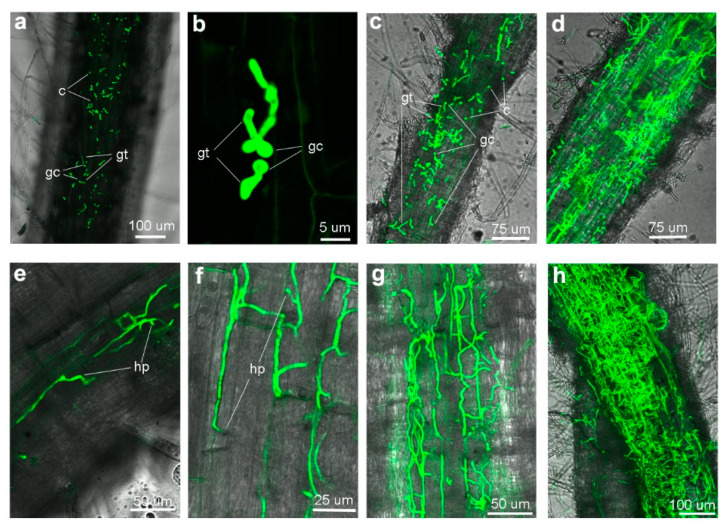
Systemic infection of eggplant roots by Vda^Sm^-GFP. (**a**) Germinated conidia on the eggplant root surface (12 hpi). c—conidium, gc—germinated conidia, gt—germ tubes; (**b**) Germ tubes emerging from one end of the conidium (12 hpi); (**c**) Vda^Sm^-GFP hyphae on the root of eggplant (24 hpi); (**d**) Hyphae covering the surface with a nonspecific growth pattern (3 dpi); (**e**) A few hyphae tightly adhered on the root surface penetrated intercellular into the epidermal cells (3 dpi). hp—hyphopodia; (**f**) Swelling hyphae were observed at the site of penetration to an adjacent cell (4 dpi); (**g**) A hyphae net within the xylem vessels was observed by 6 dpi; (**h**) Hyphae filled the root xylem vessels by 8 dpi.

**Figure 4 jof-09-00484-f004:**
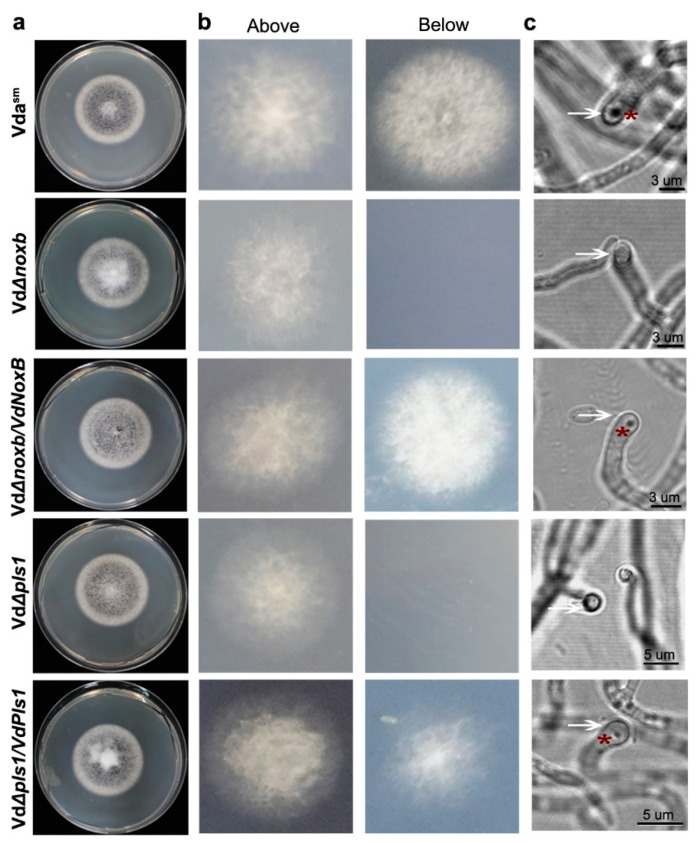
The penetration abilities assay on the cellophane membrane. (**a**) Colony morphology of wild-type Vda^Sm^, VdΔ*noxb* and VdΔ*pls1* mutants, and VdΔ*noxb/VdNoxB* and VdΔ*pls1/VdPls1* on PDA plates; (**b**) Colonies grown on the cellophane membrane (Above) and MM (Below). Vda^Sm^ grew on the MM after penetration from the cellophane membrane. Mutant strains lost penetration ability, and complemented strains restored penetration ability; (**c**) Vda^Sm^ and complemented strains developed hyphopodia with penetration pegs. The hyphopodia are indicated by the arrow, and the penetration peg is indicated by asterisks.

**Figure 5 jof-09-00484-f005:**
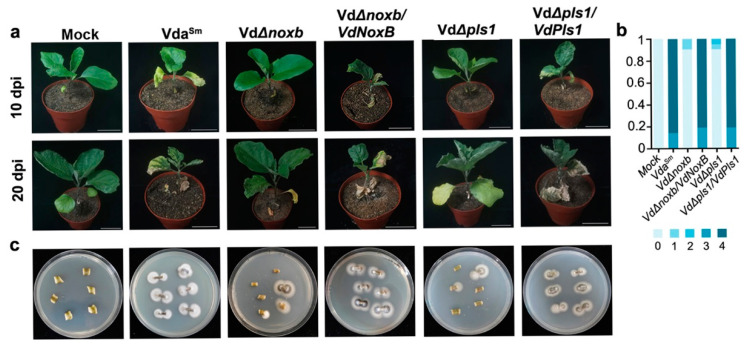
Disease symptoms of eggplants infected with Vda^Sm^, mutant strains VdΔ*noxb* and VdΔ*pls1*, and complemented strains. (**a**) Photographs were taken at 10 dpi and 20 dpi. Bar = 8 cm. Similar results were obtained from 20 infected plants for each strain, and representative photographs are shown. (**b**) Disease grade of infected plants by various strains. 21 infected plants for each strain were counted; (**c**) Isolation of diverse strains from the stems of infected plants.

**Figure 6 jof-09-00484-f006:**
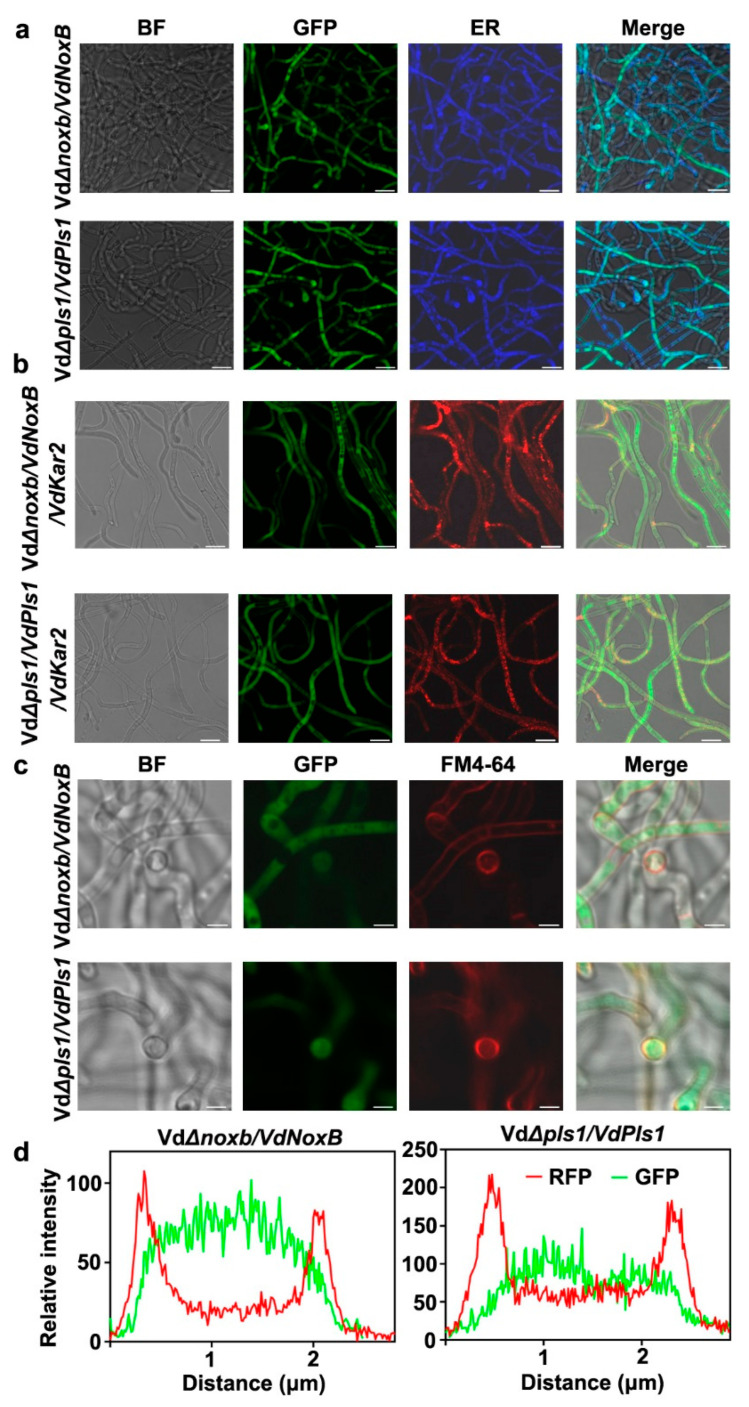
VdNoxB and VdPls1 were highly expressed in hyphopodium and localized at the base of hyphopodium. (**a**) Localization of VdNoxB and VdPls1 in the ER. ER was stained with ER-Tracker Blue-White DPX. Bar = 10 μm; (**b**) Localization of VdNoxB-GFP, VdPls1-GFP and VdKar2RFP. Bar = 10 μm; (**c**) Localization of VdNoxB and VdPls1 with the membrane of penetration pegs. The plasma membrane was stained using FM4-64, Bar = 2.5 μm. (**d**) Linescan graphs showing the relative intensity of RFP and GFP at the base of hyphopodium.

**Figure 7 jof-09-00484-f007:**
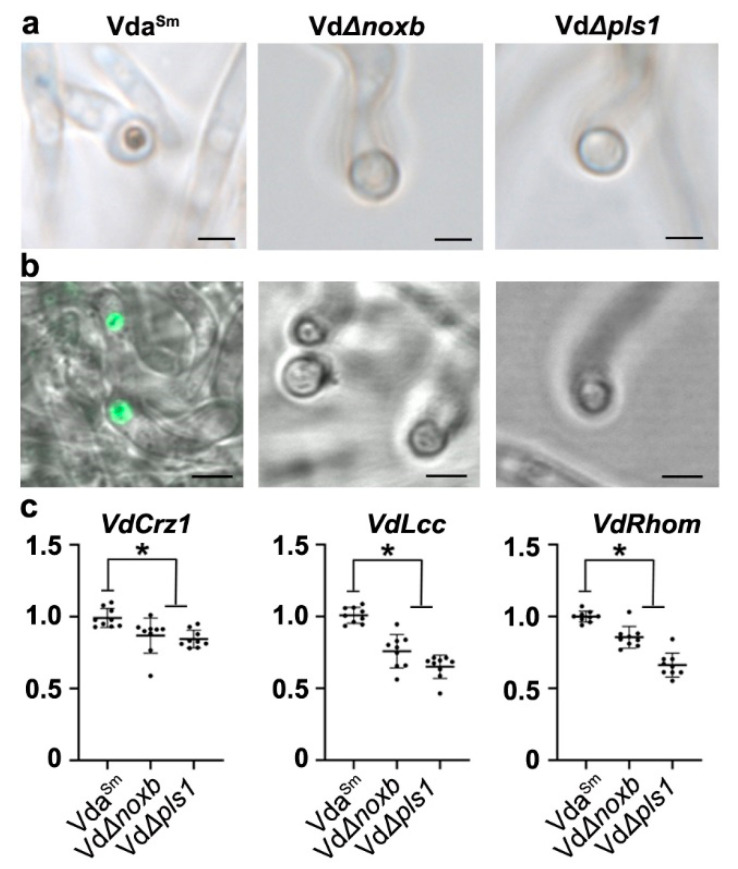
VdNoxB/VdPls1-dependent ROS burst affected VdCrz1 signaling by Ca^2+^ elevation. (**a**) CLSM observation of ROS accumulation at the base of hyphopodia in Vda^Sm^ but not in mutant strains. Hyphae grown on the cellophane membrane were stained with DAB. Bar = 10 μm; (**b**) Detection of Ca^2+^ elevation in hyphopodia with Fluo-4 AM. A tip-high gradient Ca^2+^ was observed in Vda^Sm^ but not in the VdΔ*noxb* and VdΔ*pls1* strains. Bar = 5 μm; (**c**) Expression analysis of VdCrz1 and its target genes in Vda^Sm^ and mutant strains. The asterisks indicate significant differences (*p* < 0.05, one-way ANOVA), and error bars show the standard deviations.

**Figure 8 jof-09-00484-f008:**
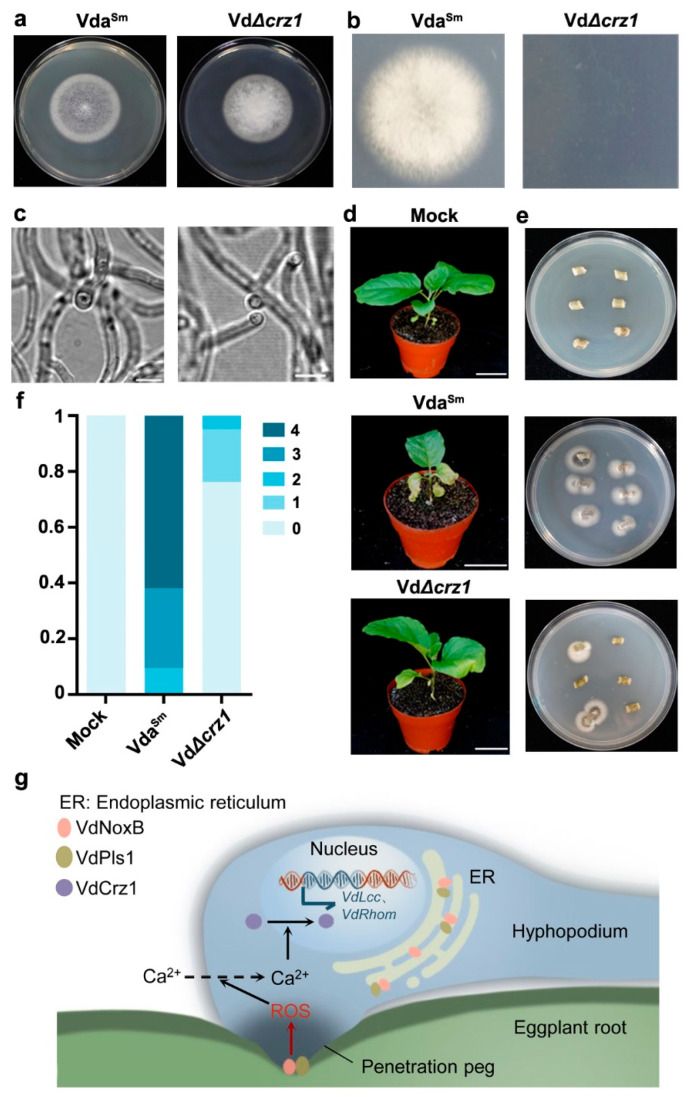
VdCrz1 signaling regulated penetration peg formation and pathogenicity of Vda^Sm^ on eggplants. (**a**) Colony morphology of wild-type Vda^Sm^ and VdΔ*crz1* mutant; (**b**) The penetration ability detection of VdΔ*crz1* on the cellophane membrane at 3 dpi; (**c**) Penetration peg was not observed in VdΔ*crz1*. Bar = 5 μm; (**d**) Disease symptoms of eggplants infected with Vda^Sm^ and VdΔ*crz1*. Similar results were obtained from 20 infected plants for each strain, and representative photographs are shown. Photographs were taken at 15 dpi. Bar = 10 cm; (**e**) Isolation of Vda^Sm^ and VdΔ*crz1* mutant strains from the stems of infected plants; (**f**) Disease grade of infected plants by Vda^Sm^ and VdΔ*crz1* mutant strains. A total of 21 infected plants for each strain were counted; (**g**) Schematic overview of VdNoxB/VdPls1-mediated ROS-Ca^2+^ signaling during penetration pef formation of Vda^Sm^ on eggplant roots.

## Data Availability

Not applicable.

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
