# Peer review of "Hyphopodium-Specific Signaling Is Required for Plant Infection by Verticillium dahliae"

_jof, 2023, doi:10.3390/jof9040484_

Round 1

Reviewer 1 Report (Previous Reviewer 3)

Dear authors,

Thank you for accepting the corrections and improving the article.

Author Response

Thanks for your careful review and constructive suggestions on our manuscript.

Reviewer 2 Report (Previous Reviewer 2)

My biggest concern regarding the revised version of the manuscript is still that I do not see a major novel contribution to the understanding of hyphopodium formation and the underlying signaling process. In summary the manuscript shows that everything described by the authors for cotton is the same for eggplant. In the revised version my minor points were nicely addressed. However, no additional data with regard to new aspects of the process or data involving factors from other publications have been added in this new version. In my opinion this limits the impact of the study as well as the interest to readers significantly. I understand that conservation of the mechanism is somehow interesting but for me this finding should be part of a manuscript combining this very basic message with novel findings and/or a more comprehensive view putting factors, which were described in other publications into a more detailed network/global picture. This would require new experiments.

I have minor points with regard to the new figures integrated:

-         - The general disease scores rank from 0-4 (e.g. in figure 5b). I did not find any information on how the score was calculated, which symptoms were measured and how these numbers would translate. Is it that 0 is no symptoms and 4 plant is dead?

-          - For the Southern blots it would be nice to mention the expected fragment sizes (e.g. included in the figure legend with respective indications also at the pictures of the blots).

Author Response

Reviewer 2

Comments and Suggestions for Authors

My biggest concern regarding the revised version of the manuscript is still that I do not see a major novel contribution to the understanding of hyphopodium formation and the underlying signaling process. In summary the manuscript shows that everything described by the authors for cotton is the same for eggplant. In the revised version my minor points were nicely addressed. However, no additional data with regard to new aspects of the process or data involving factors from other publications have been added in this new version. In my opinion this limits the impact of the study as well as the interest to readers significantly. I understand that conservation of the mechanism is somehow interesting but for me this finding should be part of a manuscript combining this very basic message with novel findings and/or a more comprehensive view putting factors, which were described in other publications into a more detailed network/global picture. This would require new experiments.
Response: Thanks for your careful review and suggestions.

Our previous study reported that VdNoxB/VdPls1-dependent ROS-Ca2+ signaling is required for cotton infection by V. dahliae. Here, we provided new evidence that VdNoxB/VdPls1-mediated signaling may be a common genetic pathway to regulate infection-related development in V. dahliae. To elucidate the molecular characterization, we would study novel factors involving in colonization process in our further study.

I have minor points with regard to the new figures integrated:

The general disease scores rank from 0-4 (e.g. in figure 5b). I did not find any information on how the score was calculated, which symptoms were measured and how these numbers would translate. Is it that 0 is no symptoms and 4 plant is dead?

Response: We have added the method of evaluating disease grades in the “Materials and Methods” section (lines 106-109).

For the Southern blots it would be nice to mention the expected fragment sizes (e.g. included in the figure legend with respective indications also at the pictures of the blots).

Response: We have marked the fragment sizes and added schematic diagrams to show the expected fragment sizes. The original electrophoretograms with DNA ladder were provided in the “original_images_for_gels” file. Thanks.

Reviewer 3 Report (Previous Reviewer 1)

1. P85 .....were collected from a field in Guan County, Langfang City, Hebei Province, China. I suggested to included GPS informaiton of disease sample collection site.

2. P87  ....The stems were cut into 1 cm slices ..., suggested change into .....The stems were cut into 1 cm slices vertically ....or ......The stems were cross-section into 1 cm slices ....  It mainly depend on how to deal with stem sample so as to get the tissue for pathogen isolation.

3. P112 .... eggplant roots (Hang No. 1 eggplant) .... suggested into .... eggplant roots (Variety: Hang No. 1)....

4. P154   we collected eggplants with Verticillium wilt ... i suggested it into: we collected disesed eggplants with the typical symptom of Verticillium wilt ....

5. P156  The hyphae that grew from eggplants were transferred.... i suggested in into : The hyphae that grow around the tissue slices of eggplant were transferred

6. P284  Taken together, these results indicated that both VdNoxB 284

and VdPls1 of Vda Sm are associated with the ER, and localized at the position of penetra- 285

tion peg emergence in VdaSm.... I am wondering ER location is very important for infection peg penetration? If it is try ,this conclusion should be discussed in discussion part.

Author Response

Reviewer 3

Comments and Suggestions for Authors

P85 .....were collected from a field in Guan County, Langfang City, Hebei Province, China. I suggested to included GPS informaiton of disease sample collection site.

Response: Added (line 90), thanks.

P87  ....The stems were cut into 1 cm slices ..., suggested change into .....The stems were cut into 1 cm slices vertically ....or ......The stems were cross-section into 1 cm slices ....  It mainly depend on how to deal with stem sample so as to get the tissue for pathogen isolation.

Response: Modified (line 91).

P112 .... eggplant roots (Hang No. 1 eggplant) .... suggested into .... eggplant roots (Variety: Hang No. 1)....

Response: Modified (line 119).

P154   we collected eggplants with Verticillium wilt ... i suggested it into: we collected disesed eggplants with the typical symptom of Verticillium wilt ....

Response: Modified (lines 162-163).

P156  The hyphae that grew from eggplants were transferred.... i suggested in into : The hyphae that grow around the tissue slices of eggplant were transferred

Response: Modified (line 165).

P284  Taken together, these results indicated that both VdNoxB and VdPls1 of Vda Sm are associated with the ER, and localized at the position of penetration peg emergence in VdaSm.... I am wondering ER location is very important for infection peg penetration? If it is try, this conclusion should be discussed in discussion part.

Response: Thanks for your careful review and suggestions. We have added this conclusion in the discussion part (lines 407-409).

Reviewer 4 Report (New Reviewer)

The authors isolated and identified the pathogen from infected eggplants collected from the field, then observed the colonization process of GFP-labeled VdaSm in the roots of eggplants, and confirmed that the formation of adhesion cells by VdaSm requires Nox/Pls-dependent signaling and depends on ROS and Ca2+ ion signals, as well as Vdâ–³crz1 signaling. This study provides a visualization of the colonization process of VdaSm and a theoretical basis for developing effective fungicides to interfere with the infection process of V. dahliae. However, there are several points worth notice:

1. To fully demonstrate the background and significance of this study, research progress on the role of ROS and Ca2+ signaling pathways in the pathogenicity is suggested to added in the “introduction part”.

2. In line 262, “3.4. VdNoxB and VdPls1 were highly expressed in hyphobodia and localized at the position of penetration peg emergence in VdaSm” should be “3.4. VdNoxB and VdPls1 were highly expressed in hyphopodia and localized at the position of penetration peg emergence in VdaSm”.

3. It is suggested to add a working model as a summary to illustrate the main findings and significance of this study.

Author Response

Reviewer 4

Comments and Suggestions for Authors

The authors isolated and identified the pathogen from infected eggplants collected from the field, then observed the colonization process of GFP-labeled VdaSm in the roots of eggplants, and confirmed that the formation of adhesion cells by VdaSm requires Nox/Pls-dependent signaling and depends on ROS and Ca2+ ion signals, as well as Vdâ–³crz1 signaling. This study provides a visualization of the colonization process of VdaSm and a theoretical basis for developing effective fungicides to interfere with the infection process of V. dahliae.

Response: Thanks for your careful review and suggestions.

However, there are several points worth notice:

To fully demonstrate the background and significance of this study, research progress on the role of ROS and Ca2+ signaling pathways in the pathogenicity is suggested to added in the “introduction part”.

Response: We added the background of ROS and Ca2+ signaling pathways in the pathogenicity in the “Introduction” section (lines 73-77).

In line 262, “3.4. VdNoxB and VdPls1 were highly expressed in hyphobodia and localized at the position of penetration peg emergence in VdaSm should be “3.4. VdNoxB and VdPls1 were highly expressed in hyphopodia and localized at the position of penetration peg emergence in VdaSm”.

Response: Corrected (line 271).

It is suggested to add a working model as a summary to illustrate the main findings and significance of this study.

Response: We have added a working model as Figure 8g. Thanks.

Round 2

Reviewer 2 Report (Previous Reviewer 2)

My minor points were addressed properly, thanks.

Nothing changed with regard to my concerns regarding novelty and interest to readers.

This manuscript is a resubmission of an earlier submission. The following is a list of the peer review reports and author responses from that submission.

Round 1

Reviewer 1 Report

This manuscript titled with Hyphopodium-specific signaling is required for plant infection by Verticillium dahliae described the infection process of V. dahliae isolated from disease eggplant and also the underlyed the signalling pathway related to the penetration process. However, i think the main line of this manuscript is not very clear. Based the published papers on infection process of V. dahliae on cotton , the molecular mechanism of the infection process of V. dahliae on cotton was described clearly by Zhao, Y. L.(2016) and Zhou TT.(2017). Due to the same pathogen, i think the infection mechanism of V.dahliae does not make big difference on different hosts. So, i think the main purpose of this manuscript does not very meaningful. 

1. Line 47: here described quite a lot information of infection process of Rbizoctonia solani, i think these information has less relation with the main topic of this manuscript, i suggested to delete this part and increased the infection process of V. dahliae on different crops, such as lettuce, potato ects.

2. Line 86: here should indicated the location information where the diseased samples were collected?And how many samples were collected and how many isolates were obtained in this study?

3. Line 90:Here need information of how to perform the isolates purification.

4. Line 112: here should indicated the variety name of eggplant which was used for inoculation

5. Line 151: reisolation needed to describe to confirm the successful inoculation

6. Line 158: here also need the identification isolate with specific primer of V. dahliae

7. Line 227:in my opinion, the knock out mutant do affect the penetration ability, but it seems do not affect the penetration structure.

8. Line 242:Here need the diseased index to indicate the pathogenicity of different knock out mutants

9. Line 270:Figure 6. VdNoxB and VdPls1 were highly expressed in hyphopodium and localized at the base of hyphopodium. I do not think the co-staining of both GFP and ER marker can give clear information for drawing the conclusion.

Reviewer 2 Report

The authors study the conservation of hyphopodium formation on plant roots and the underlying signaling process. The authors refer to their previous work in cotton, where they observed hyphopodium formation and signaling through NoxB, Pls1 and Crz1 and now investigate the infection of eggplants. Overall, the manuscript is well written. Most results (despite the isolation of V. dahliae from eggplants) appear to me as a repetition of experiments that have been carried out by the authors in cotton. The outcome, hyphopodium formation and underlying signaling is the same for the two isolates and their host plants seems to be not surprising but rather expected. The authors also missed the opportunity to possibly extend their studies to the factors that have been studied in other publications (e.g. “Cellophane surface-induced gene, VdCSIN1, regulates hyphopodium formation and pathogenesis via cAMP-mediated signalling in Verticillium dahliae“) or also new factors but restricted their studies to what their lab has published before. For me, the manuscript overall lacks novelty and I do not see how it can contribute to “help to develop more effective fungicides to protect crops from V. dahliae infection” (as stated by the authors) as the molecular mechanisms have been published before by the group. It would have been interesting to investigate the relevance of the NoxB/Pls1/Crz1 (and further hyphopodium factors) signaling in Verticillium-pathogen interactions, where the fungus is believed to form appressoria to see if the mentioned proteins also play a function in this process or if the regulation is completely different.

Other points that I noticed:

-          To me it is not clear where the border is between the “hyphal swelling” that has been observed in several Verticillium-plant interaction studies (some of them mentioned in the introduction) and the formation of a “real” hyphopodium that the authors describe. If I look at the different micrographs, they look quite similar to me. Is it likely that also this hyphal swelling is also hyphopodium formation? How is the differentiation?

-          The authors constructed several deletion and complementation strains. In figure S2/S3 I can see that they were verified by PCR. This ensures the replacement of the respective genes but it does not show if there was an additional off-target insertion of the cassette. For this Southern blots are required.

-          Figure S1: The figure legend is very short. I also do not see any growth rate (quantification of growth during a specific time interval) in the figure but just a picture of the colony on plate.

Minor

-          Figure 3: I guess “hp” is used to mark hyphopodia? It is not explained in the figure legend.

-          Line 299: Would be nice to shortly say what kind of proteins these potential target genes encode.

-          Line 215: If MM is used for “minimal medium” the medium in the text can be removed (same is true for line 229, 230).

-          Line 34: Appressoria have… (as it is plural)

-          Line 47: Rhizoctonia (instead of Rbizoctonia)

-          Line 162: wilt not italic

Reviewer 3 Report

Dear Authors,

I would like to see a separate heading with the Conclusions, please add it.

Please add a DOI number for all references, if available.

Also, some language and grammar improvements are needed. Examples are included in the comments below.

Comments:

L11 - Please delete "elaborate"

L14 - Please change "roots," to "roots"

L15 - Please change "while develops appressoria, that are typically associated with leaf infection, on lettuce and fiber flax" to "while developing appressoria, that are typically associated with leaf infection on lettuce and fiber flax"

L31 - Please delete "elaborate"

L47 - Please change "Rbizoctonia solani" to "Rhizoctonia solani"

L86 - Please add "symptoms" after the word "wilt"

L162 - Please change "wilt" to "wilt"
